# Efficacy of NAMPT Inhibitors in Pancreatic Cancer After Stratification by MAP17 (PDZK1IP1) Levels

**DOI:** 10.3390/cancers17152575

**Published:** 2025-08-05

**Authors:** Eva M. Verdugo-Sivianes, Julia Martínez-Pérez, Lola E Navas, Carmen Sáez, Amancio Carnero

**Affiliations:** 1Instituto de Biomedicina de Sevilla (IBIS), Hospital Universitario Virgen del Rocío (HUVR), Consejo Superior de Investigaciones Científicas, Universidad de Sevilla, 41013 Seville, Spain; everdugo-ibis@us.es (E.M.V.-S.); julia.martinez.perez.sspa@juntadeandalucia.es (J.M.-P.); dnavas-ibis@us.es (L.E.N.); csaez1@us.es (C.S.); 2CIBERONC, Instituto de Salud Carlos III, 28029 Madrid, Spain; 3Departamento de Ciencias de la Salud y Biomédicas, Facultad de Ciencias de la Salud, Universidad Loyola Andalucía, Avda. de las Universidades s/n, Dos Hermanas, 41704 Sevilla, Spain; 4Oncology Department, Hospital Universitario Virgen del Rocío, 41013 Seville, Spain; 5Pathology Department, Hospital Universitario Virgen del Rocío, 41013 Seville, Spain

**Keywords:** pancreatic cancer, prognosis, MAP17, NAD, biomarkers, chemotherapy

## Abstract

Pancreatic cancer (PC) is the seventh leading cause of cancer-related deaths globally, marked by a high mortality due to its aggressive nature and late diagnosis. Although surgical resection is the only curative option, most cases are diagnosed too late for surgery. This study identifies potential treatment vulnerabilities by focusing on MAP17 (PDZK1IP1), whose high mRNA expression correlates with poor prognosis and is found exclusively in tumor cells. In 2D cultures, MAP17-expressing cells responded better to gemcitabine and 5-fluorouracil. However, in vivo xenograft models showed resistance to all treatments. These cells also had elevated NAD levels, which could be depleted using NAMPT inhibitors. Indeed, depletion of NAD seems to resensitize MAP17-expressing tumors to chemotherapy. Thus, MAP17 may serve as a prognostic marker and therapeutic target. Combining NAMPT inhibitors with standard therapies could enhance treatment efficacy in patients with high MAP17 expression.

## 1. Introduction

Pancreatic adenocarcinoma (PC) is the seventh leading cause of cancer-related deaths worldwide [1,2]. Although its incidence in the population is not high, its aggressive nature results in high mortality, with only about 12% of patients surviving five years post-diagnosis [2,3,4]. Due to the limitations of current diagnostic techniques, the majority of patients are diagnosed at either a locally advanced or metastatic stage. Therefore, surgical resection is the only potentially curative treatment in a localized stage. However, even for those undergoing complete resection, the prognosis is poor, with high rates of systemic (80%) and local (20%) recurrence within two years after surgery. Despite recent treatment advances, prognosis has not improved over the past two decades, but survival rates have increased from 5% to 12% due to a better surgical approach and the use of neoadjuvant chemo-radiotherapy [1,5]. Given that post-surgical adjuvant chemotherapy has been shown to improve survival, it is routinely recommended. Moreover, neoadjuvant systemic therapy with or without radiation is an accepted treatment approach for resectable and borderline resectable disease [6,7]. On the other hand, the management of metastatic pancreatic cancer focuses on symptom control, management of jaundice, palliative systemic chemotherapy, improving quality of life and prolonging survival. Multi-agent chemotherapy has improved survival in the palliative setting although an optimal regimen has not been fully established and no targeted agents are available in this context either [8].

MAP17, coded by the *PDZK1IP1* gene, is a small nonglycosylated protein of 17 kDa located on the plasma membrane and in the Golgi apparatus [8]. This protein contains two transmembrane regions and a hydrophobic C-terminus encoding a PDZ-binding domain, allowing interaction with PDZ Kidney Protein 1 (PDZK1) and PDZK1-associated proteins [9]. MAP17 is known to contribute as a cargo protein to the cell membrane localization of many transporters through PDZK1-bound sodium-glucose linked transporters. Normally, MAP17 is expressed only in kidney proximal tubule cells, and its physiological role is not fully understood [8,10,11,12]. However, overexpression of MAP17 has been described in many human carcinomas, playing an oncogenic role [9,13]. The overexpression of MAP17 confers a proliferative advantage to the cells by decreasing the apoptotic sensitivity and increasing the migratory capacity compared to cells without MAP17 expression. However, the exact mechanism underlying the increased tumor progression caused by MAP17 overexpression is not fully clarified. For instance, MAP17 increases cellular reactive oxygen species (ROS) levels [9], promoting cell proliferation and growth. Additionally, MAP17 overexpression activates the Notch pathway in tumor cells in vitro, increasing the stem cell pool [14]. In addition, a correlation between MAP17 and an inflammatory phenotype in some tumors has also been described. Chronic inflammation can lead to neoplastic transformation and tumor progression, suggesting that MAP17 plays a role in carcinogenesis by regulating the immune environment [15,16]. In line with this, MAP17 activation is associated with a more aggressive phenotype and worse prognosis in several cancers, including lung, rectal, cervical, and laryngeal cancer [13,17,18,19]. Furthermore, elevated MAP17 levels are linked to a good response to platinum-based compounds in cervical and laryngeal carcinoma, and increased sensitivity to bortezomib in sarcomas, breast cancer, and lung cancer [13]. In cervical cancer, high MAP17 levels combined with cisplatin and radiotherapy lead to improved patient survival [13]. Furthermore, MAP17 expression can also predict the response to chemoradiotherapy in rectal cancer [13]. Therefore, MAP17 is not only a biomarker for malignancy and tumor progression but may also be correlated with therapy response.

Based on previous studies, our project aimed to evaluate MAP17 expression in pancreatic cancer and assess its potential as a prognostic and/or predictive biomarker. We also explored whether the response to various antitumor agents depends on MAP17 expression and whether it is possible to resensitize these cells to chemotherapy. This study also aimed to investigate if MAP17-expressing cells have a higher NAD pool, and if the depletion of this pool by Nicotinamide phosphoribosyltransferase (NAMPT) inhibitors could sensitize cells to classic chemotherapy in pancreatic tumor xenografts.

## 2. Materials and Methods

### 2.1. HUVR-IBIS Clinical Cohort Description

From 2000 to 2018, patients diagnosed at our institution, the Instituto de Biomedicina de Sevilla (IBIS) and Hospital Universitario Virgen del Rocío (HUVR), Seville, Spain, who met the following inclusion criteria were selected for the present study: (1) histologically confirmed diagnosis of primary pancreatic cancer, (2) surgical resection with curative intent, and (3) tumor tissue available for additional immunohistochemical assays. Medical records were retrospectively reviewed according to a previously designed protocol to record the most relevant demographic, clinical and pathological features. The study protocol was approved by the institutional Ethics Committee (CEI_1953-N-18) and met national guidelines for noninterventional studies dealing with human subjects. Statistical analyses were performed using IBM SPSS version 20.0 for Windows. Descriptive statistics were used to characterize the most relevant clinical parameters. A total of 97 patients were included in this study. At the time of analysis, 80.4% of recurrences had been documented, and the median follow-up was 20.4 months (6.7–166.2). Details of the baseline demographic and tumor characteristics are presented in Appendix A. Adjuvant therapy was administered to 66% of the patients; 58.8% of patients received gemcitabine-based regimens, and 6.2% of patients received fluoropyrimidine-based therapy (5-fluorouracil (5-FU) or capecitabine in monotherapy in the majority of cases or FOLFIRINOX in a few patients (*N* = 3)).

### 2.2. Immunohistochemical Analysis of Protein Expression

Archived, formalin-fixed, paraffin-embedded tissue sections from 97 pancreatic tumors provided by the Biobank of the Hospital Virgen del Rocio (Seville, Spain) were selected. Seven tissue microarrays (TMAs) were constructed with 2 paired samples (tumor sample and adjacent normal tissue) of each patient. Histological characterization of all the samples was performed by hematoxylin and eosin staining, followed by immunohistochemistry (IHQ) analysis of TMAs. All histological slides were independently assessed by two pathologists with more than 20 years of experience in the field. Each tumor was assessed using two distinct tissue sections that covered a representative part of the tumor, with two independent and blinded evaluations to ensure reliability and minimize observer bias. In cases of discrepancy, the most restrictive evaluation was adopted to ensure consistency and minimize overestimation.

The tissue microarray was composed of duplicate 5-μm-diameter cores. Briefly, tissue sections were dewaxed, rehydrated and immediately immersed in 3% H_2_O_2_ aqueous solution for 30 min to exhaust endogenous peroxidase. Heat-induced epitope retrieval was performed with 1 mM EDTA (pH 9.0) in a microwave oven. Then, sections were incubated overnight at 4 °C with mouse primary antibody against human MAP17 (1:30 plus linker mouse). Peroxidase-labeled secondary antibody and 3,3-diaminobenzidine were applied according to the manufacturer’s protocol (EnVision, Dako, Agilent Technology, Glostrup, Denmark). Tissues were then counterstained with hematoxylin. The immunostained area was measured by observation according to the anatomopathological experience of the pathologists. Tumor samples were classified according to the number of stained cells and signal intensity into 4 categories: 0 (no staining), 1 (<25% cells were stained, and cytoplasm immunostaining intensity was weak), 2 (>25% cells were stained, and the immunostaining intensity was moderate) or 3 (>50% of the cells were stained, and the immunostaining intensity was high). Patients were classified on the basis of the median MAP17 expression levels in both groups of assessed samples: (1) low MAP17 expression, MAP17 signal intensity ≤ 0.75 and (2) high MAP17 expression, MAP17 signal intensity > 0.75. The optimal cutoff point was obtained on the basis of the ROC, which was assessed to determine the most relevant dichotomous variables using IBM SPSS version 20.0.

### 2.3. Public Databases of Clinical Samples

Data from publicly available clinical and genomic information were obtained from the R2 Genomics analysis and visualization platform (https://hgserver1.amc.nl/cgi-bin/r2/main.cgi; accessed on January 2025), the TCGA Research Network (https://cancergenome.nih.gov/; accessed on November 2022) and the TISCH resource (http://tisch1.comp-genomics.org/; accessed on November 2022) [20]. We analyzed *MAP17*, *NAMPT* and Nicotinate phosphoribosyltransferase (*NAPRT*) expression levels in tumor and nontumor pancreatic samples obtained from these databases. The statistical significance of tumor versus nontumoral samples was assessed (*p* < 0.05). Kaplan–Meier plots showing patient survival were generated using databases with available survival data via the scan method, through which the optimum survival cut-off was assessed based on statistical analyses (log-rank test), and the most significant expression cut-off factor was determined. In the correlation of MAP17 with the enzymes related to NAD biosynthesis and consumption, we selected those that correlate with a Pearson correlation coefficient greater than 0.25 and a *p*-value < 0.05 and the heat map was generated by Euclidean distance.

### 2.4. Cell Culture

PANC-1 and HPAF-II cell lines were obtained from the ECACC commercial repository. No further authentication was performed by the authors. Cells were tested for mycoplasma and found to be negative. Both cell lines were maintained in DMEM (AQmedia; Sigma, St. Louis, MO, USA) supplemented with 10% fetal bovine serum (FBS) (Gibco, Paisley, UK), penicillin, streptomycin and fungizone (Sigma). HPAF-II cells were also supplemented with nonessential amino acids and sodium pyruvate.

### 2.5. Transfection and Plasmids

Subconfluent cells were transfected with TransIT-X2 reagent (Mirus, Madison, WI, USA) according to the manufacturer’s instructions. Forty-eight hours later, the cells were seeded in 10-cm plates with medium containing the appropriate level of selection drug (0.5–1.5 μg/mL puromycin). The cells were transfected with the pBabe-empty plasmid vector (EV in the text) and pBabe-MAP17 as previously described [15].

### 2.6. RT–qPCR

Total RNA from cell lines was extracted and purified using a ReliaPrepTM RNA Tissue Miniprep System (Promega, Madison, WI, USA), and reverse transcription was performed with 3 µg of mRNA using a High Capacity cDNA Reverse Transcription kit (Life Technologies, Carlsbad, CA, USA) according to the manufacturer’s instructions. The PCR mixture (10 µL) contained 2 µL of the reverse transcriptase reaction solution diluted 1:10 with 2.5 µL of water and to which 5 µL of GoTaqR Probe qPCR Master Mix (Promega) and 0.5 µL of the appropriate TaqMan Assay (20X) (Applied Biosystems, Foster City, CA, USA) were added. We used the following probes: GAPDH (Hs03929097_g1) as an endogenous control and MAP17 (Hs00906696_m1).

### 2.7. Protein Isolation and Western Blot Analysis

Western blotting was performed as previously described. The blots were incubated with the following primary antibodies: anti-MAP17 (Millipore MABC522, Billerica, MA, USA), anti-NAMPT (Bethyl A300-779A, Waltham, MA, USA), anti-NAPRT (Sigma SAB1400768), anti-SIRT1 (Cell Signaling 2310, Danvers, MA, USA) and anti-PARP (Cell Signaling 9532) antibodies with anti-HSP70 (Abcam ab45133, Cambridge, UK) and anti-α-tubulin (Sigma T9026) antibodies used as the loading controls. Horseradish peroxidase-labeled rabbit anti-mouse (Abcam ab97046) and goat anti-rabbit (Abcam ab97051) secondary antibodies were also used. Protein levels were measured using an ECL detection system (Amersham Biosciences, GE Healthcare, Buckinghamshire, UK) and Bio–Rad Chemidoc Touch (Hercules, CA, USA).

### 2.8. Growth Curve

A total of 5 × 10^3^ cells were seeded in 6-well plates in triplicate. At 24 h (day 0), cells were fixed with 0.5% glutaraldehyde (Sigma) and every 48h a curve point was fixed up to 10 days. Once all the points were collected, plates were stained with 1% violet crystal (Sigma). Then, the violet crystal was dissolved with 20% acetic acid (AppliChem, Darmstadt, Germany) and the relative number of cells was quantified by measuring the absorbance of the violet crystal at 595 nm by using an absorbance reader (Biorad). The values were represented referring to day 0.

### 2.9. Clonogenic Assay

A total of 1 × 10^3^ cells were plated in 10 cm plates in triplicate. Cells were fixed with 0.5% glutaraldehyde and stained with 1% violet crystal after 15 days. The number of colonies was counted and types of clones classified based on the following classification: holoclone (more compact and rounded clone with a higher percentage of cancer stem cells, (CSCs)), meroclone (clone with a more irregular cell arrangement at the edges and a lower proportion of CSCs), or paraclone (greater separation of cells from each other, due to a greater degree of differentiation and lower percentage of CSCs) [21,22,23]. The morphology of each clone was observed and photos of each type of clone were taken under an inverted microscope (Olympus IX-71, Olympus, Tokyo, Japan).

### 2.10. Tumorsphere Assay

A total of 1 × 10^3^ cells were seeded in triplicate in 24-well ultralow attachment plates (Costar, Corning, NY, USA) containing 1 mL of MammoCult basal medium (Stem Cell Technologies, Vancour, BC, Canada) supplemented with 10% MammoCult proliferation supplement, 4 μg/mL heparin, 0.48 μg/mL hydrocortisone, penicillin and streptomycin. After 5–10 days, depending on the cell line, the number of primary tumorspheres formed (spherical and solid structures, where individual cells are not easily distinguished and different from single or aggregated cells) was measured using an inverted microscope (Olympus IX-71).

### 2.11. Cytotoxicity Assay

Cytotoxicity studies were performed as described previously. Briefly, 15,000 cells/well were seeded in 96-well plates and allowed to attach and grow for 24 h before treatment. The drugs (gemcitabine, 5-FU, cisplatin, docetaxel, bortezomib, ixazomib, erlotinib, and osimertinib) were weighed, diluted in sterilized deionized water or DMSO and then applied to the 96-well master plate at decreasing concentrations. In the case of the combined treatment, we applied one drug at decreasing concentrations and then we applied the second drug at two different doses that correspond to the IC50 and IC30 for this drug (named D1 and D2, respectively). Treatment was applied for 96 h, and then, the cells were fixed with 0.5% glutaraldehyde and stained with 1% crystal violet. The number of proliferating cells was measured by dissolving crystal violet in 20% acetic acid and measuring the absorbance at 595 nm with an absorbance reader. The concentration necessary to induce the death of 50% of the cells (IC50) was calculated using GraphPad Prism 6 software.

### 2.12. Fluorescence-Activated Cell Sorting (FACS) Analysis

For FACS analysis, 1 × 10^6^ cells were trypsinized and suspended in 125 μL of PBS containing 2% FBS and 5 mM EDTA. The cells were blocked with 12.5 μL of human blocking reagent (Miltenyi Biotec, Bergisch Gladbach, Germany) for 10 min at 4 °C. Then, the cells were incubated with 2 μL of anti-CD133-PE (Miltenyi Biotec #130-113-670) for 30 min at 4 °C. After washing twice with PBS-FBS-EDTA, the cells were suspended in 500 μL of the same buffer and analyzed by FACS with a FACS Canto II cytometer (BD Biosciences, Franklin Lakes, NJ, USA). Experiments established with triplicate samples were repeated independently at least three times.

### 2.13. Metabolism Assays

NAD total cellular pools were quantified following the protocol detailed in the article Zhu et al., 2012 [24] and adapted by the research group of Dr. Lindsay Wu. Cell media were removed, plates were washed in cold PBS and cells were scraped down in a buffer containing 50 mM Tris HCl, 0.1% Triton X-100 and 10 mM nicotinamide, to inhibit the activity of NAD-degrading enzymes. Cells were homogenized by sonication for 5 s, and samples were centrifuged at 7000× *g* for 5 min at 4 degrees. Aliquots were taken for later protein assay, and samples were then passed through 10 kDa amicon filters at 14,000× *g*, 30 min at 4 degrees to remove proteins from the sample. Flow-throughs were separated into reactions for NAD total (NAD+ and NADH) and NAD+. To one of these samples, HCl was added to 10 mM and sample heated at 70 °C for 30 min to degrade NADH, leaving behind NAD+ alone. For both aliquots, 25 μL sample was then added to 100 μL ADH cycling mix (0.2 mg/mL alcohol dehydrogenase enzyme, 2% ethanol, 100 mM Tris pH 8.5). Samples were allowed to cycle for 10 min at room temperature, followed by 50 μL addition of an MTT/PMS solution (0.1 mM phenazine methosulfate, 0.8 mM 3-(4,5-Dimethylthiazol-2-yl)-2,5-diphenyltetrazolium bromide), 100 mM Tris–HCl pH 8.5). Plates were then incubated for 15 min and absorbance was measured at 570 nM. NAD concentrations were extrapolated from a standard curve and normalized to protein concentrations determined by BCA protein assay.

### 2.14. Xenografts in Nude Mice

Tumorigenicity was assayed after the subcutaneous injection of 5 × 10^6^ PANC-1 cells into the right flanks of 4-week-old female athymic nude mice. Cells had been suspended in 50 μL of Matrigel (Corning) prior to injection. Animals were examined weekly. After 70 days, the mice were sacrificed, and tumors were extracted and stored at −80 °C. Mice inoculated with the PANC-1 cell line were treated with a 2 mg/kg dose of cisplatin (2 doses/week), 20 mg/kg dose of 5-FU (2 doses/week), 25 mg/kg dose of gemcitabine (2 doses/week), 200 mg/kg dose of GMX1778 (1 dose/week) or 30 mg/kg dose of GNE617 (5 doses per week). The drugs cisplatin, 5-FU and gemcitabine were obtained from the HUVR pharmacy and were freshly prepared and administered by intraperitoneal injection. GMX1778 (ForLab, Indra Sistemas, Madrid, Spain) and GNE671 (ForLab) were administered by oral gavage. The treatments were applied one week after the injection of the cells. All mice received treatment for 3 weeks. The animals were monitored weekly for signs of distress, and we did not observe signs of toxicity. Tumor volume (mm^3^) was measured using calipers. All animal experiments were performed according to the experimental protocol approved by the IBIS and HUVR Institutional Animal Care and Use Committee (CEI_1953-N-18).

### 2.15. Statistical Analysis

Statistical analyses of the experiments were performed using GraphPad Prism (6.01 for Windows). Control samples (non-MAP17-overexpressing) and MAP17-overexpressing samples were compared using an unpaired Student’s *t* test or Student’s *t* test with Welch’s correction as appropriate. Experiments with triplicate samples were independently performed at least three times. *p* values less than 0.05 were considered to be statistically significant. They are presented according to the following classification: *p* < 0.05 (*), *p* < 0.01 (**), and *p* < 0.001 (***).

## 3. Results

### 3.1. MAP17 Is Upregulated in Pancreatic Tumors

Previously, in our laboratory we have found that the level of expression of MAP17, either mRNA or protein, could be a good reference for stratifing patients according to prognosis [13]. To assess MAP17 expression in pancreatic cancer, we analyzed its expression in a TCGA cohort of pancreatic adenocarcinoma patients which included both normal and tumor samples and we found that MAP17 levels were significantly elevated in tumor samples compared to normal samples (Figure 1A). In addition, we examined three public pancreatic cancer databases, the GSE62165, GSE62452 and GSE183795 datasets [25,26,27], and again MAP17 mRNA levels were significantly higher in tumor samples than in normal samples (Figure 1B). We also analyzed a protein database, LinkedOmics [28], and we found that the protein levels of MAP17 were also significantly higher in the tumor samples in comparison with the normal samples (Figure 1C). These data were corroborated at the protein level by measuring MAP17 in nontumor pancreatic tissue and adenocarcinoma tumor samples from a HUVR-IBIS cohort via immunohistochemistry (IHQ), revealing increased MAP17 protein expression in tumor samples compared to normal stroma or nontumor cells (Figure 1D,E).

In the tumor samples, MAP17 was observed in 57 (58.8%) patients. Among these, IHQ staining intensity was weak (1) in 21 (21.6%) samples, moderate (2) in 29 (29.9%) samples, and intense (3) in 7 (7.2%) samples. High MAP17 protein expression was defined as an IHQ intensity value greater than 0.75, while low expression was defined as an intensity value of 0.75 or lower (Figure 1E), similar to the cut-point used in other tumor types [9,13]. Under these criteria, 60 (61.9%) patients exhibited low MAP17 expression, and 37 (38.1%) exhibited high expression (Figure 1F). Single-cell analysis of MAP17 expression among different cell lineages in pancreatic tumors and the microenvironment showed significantly high levels only in malignant cells (Figure 1G,H).

To evaluate the relationship between increased MAP17 expression and patient survival, we analyzed overall survival using data from the TCGA, GSE62452 and GSE183795 datasets. In the three datasets, patients with high MAP17 expression had decreased survival compared to those with low MAP17 expression, but this result was statistically significant only in TCGA (Figure 1I,J).

Overall, these results indicate that MAP17 expression is increased in a significant proportion of pancreatic tumors, correlating with decreased survival probability.

### 3.2. The Overexpression of MAP17 Enhances the Growth Properties and Stemness Capability of PANC-1 Cells

MAP17 enhances the tumorigenicity of cells in other tumors [13]. To study the specific effect of MAP17 in pancreatic cancer, we overexpressed MAP17 in two different pancreatic cancer cell lines (PANC-1 and HPAF-II) using an empty vector as a negative control. Thus, cells transfected with MAP17 are referred to as MAP17-overexpressing cells, while those transfected with the empty vector are referred to as control cells. The overexpression of MAP17 was validated at the protein level by Western blotting (Figure 2A and Appendix A) and at the mRNA level by RT-qPCR (Figure 2B). When seeded at low density, MAP17-expressing PANC-1 and HPAF-II cells formed more colonies than control cells, but this increase was statistically significant only in PANC-1 cells (Figure 2C). Additionally, overexpression of MAP17 led to faster proliferation of PANC-1 cells compared to control cells (Figure 2D).

Metastasis and chemotherapy resistance are suggested to be related to cancer stem cell physiology [29,30]. To explore the effect of MAP17 overexpression on the stemness capacity of these cell lines, we performed a clonability assay to measure clone phenotypes formed after seeding the cells at low density [21,22,23]. We observed that cells overexpressing MAP17 in both cell lines formed more holoclones (colonies enriched in cancer stem cells (CSCs)), but again this increase was statistically significant only in PANC-1 cells. In addition, cells overexpressing MAP17 formed fewer paraclones (colonies enriched in mature non-stem cells) than control cells in both cell lines, indicating an enhanced regenerative capability in culture (Figure 2E). Moreover, MAP17-expressing cells, especially HPAF-II cells, formed a higher percentage of tumorspheres (Figure 2F), suggesting an increased stemness capability. An increased proportion of CD133+ cells was identified in both cell lines, particularly in the HPAF-II cell line, with MAP17 overexpression, although this result was not statistically significant (Figure 2G). These results suggest that MAP17 expression partially enhances the growth properties and stemness capability of PANC-1 cells, potentially increasing the cancer stem cell (CSC) pool.

Next, when control and MAP17-expressing PANC-1 and HPAF-II cells were injected into nude mice, PANC-1 cells overexpressing MAP17 formed larger tumors that grew faster than those formed by control cells in the xenograft mice. The growth pattern of HPAF-II cell xenograft tumors was similar or even slow compared to that of their respective control cells (Figure 2H,I and Appendix A) perhaps reflecting an increase in non-fast-cycling CSCs.

### 3.3. MAP17 Is a Potential Biomarker of Antitumoral Response in Pancreatic Cancer

Patients with pancreatic tumors exhibiting high levels of MAP17 had a poor prognosis. Therefore, we aimed to evaluate the potential role of MAP17 as a predictor of response or resistance to antitumoral therapies, including clinically used drugs. To explore the effect of MAP17 on the response to these drugs in vitro, we determined the drug concentration that leads to 50% cell growth inhibition (IC50) in both pancreatic tumor cell lines (PANC-1 and HPAF-II) and correlated these data with MAP17 levels (Table 1).

We found that high MAP17 expression significantly correlated with increased sensitivity to gemcitabine and fluorouracil (5-FU) in vitro 2D cultures in both cell lines, but only in the PANC-1 cell line did it predict sensitivity to cisplatin. Conversely, MAP17 overexpression induced chemoresistance to docetaxel in the PANC-1 cell line. Previously, we demonstrated that the proteasome inhibitor bortezomib was effective in high-MAP17-expressing tumors from various origins, including sarcomas, breast cancer, and lung cancer tumors [13,17,18,19]. To study the efficacy of this drug in pancreatic tumors, we tested bortezomib and ixazomib, another proteasome inhibitor, in our pancreatic tumor cells in 2D cultures. We found a trend indicating that increased MAP17 expression was closely associated with resistance to bortezomib and ixazomib (Table 1).

Then, we explored the effect of some of those treatments in vivo using PANC-1 control and PANC-1 MAP17-expressing tumors. In contrast to the results obtained in vitro in 2D cultures, tumors with high levels of MAP17 did not respond to any treatment in vivo. While parental cells showed some response to cisplatin or 5-FU treatment, MAP17-expressing cells were clearly resistant to all treatments (Figure 3A–C).

Thus, we sought to identify a possible combination of therapies to resensitize currently used treatments, particularly in resistant MAP17-expressing tumors, which account for as many as 40–50% of pancreatic tumors.

### 3.4. Alterations in the Expression of Genes Related to NAD Metabolism That Correlated with MAP17 in Pancreatic Tumors

NAD is an essential metabolite involved in many cellular processes, and the NAMPT enzyme, a NAD phosphoribosyltransferase, is a key regulator of the salvage pathway in NAD biosynthesis [31]. NAMPT and NAPRT levels are significantly increased in pancreatic cancer tumors (Figure 4A–E), and its increased expression correlated with worse prognosis (Figure 4F–H). Inhibition of NAMPT has been reported to sensitize pancreatic cancer cells to gemcitabine chemotherapy by decreasing NAD levels and suppressing glycolytic activity [32,33,34,35]. Therefore, targeting NAD represents a potential novel therapy for pancreatic cancer.

First, to confirm a functional relationship between MAP17 and this pathway, we measured total NAD levels in both cell lines and we found that MAP17-overexpressing cells had higher total NAD levels than their respective control cells (Figure 5A).

In previous studies, we found that cells expressing high levels of MAP17 exhibit an increased metabolic rate, leading to excessive ROS production and correlating with increased levels of membrane glucose transporters [9,12]. Since NAD is a cofactor required for glucose metabolism, we analyzed the correlation between MAP17 and different enzymes involved in NAD biosynthesis, such as the previously mentioned NAMPT and NAPRT and the three nicotinamide nucleotide adenylyltransferases (NMNAT1-3); some NAD-consuming enzymes, such as sirtuins (SIRT1-6), poly (ADP-Ribose) polymerases (PARP1-2), CD38, CD73 and CD157; and other NAD metabolite transporters such as Cx43 [36]. We correlated MAP17 with these 17 proteins in 12 different pancreatic cancer databases and we found that in general, NAMPT and NAPRT, the two main generators of NAD, were increased in most databases correlating with higher levels of MAP17 (Figure 5B). However, NAD-consuming enzymes such as Sirtuins 1,2,3 and 4, and more importantly PARPs, were diminishing, in most databases (Figure 5B).

Figure 5C,D shows example of the positive correlation between NAMPT and NAPRT with MAP17 in three of the six databases (TCGA, GSE93326 and GSE208732 or GSE21501) with a Pearson correlation coefficient higher than 0.4. PARP1 and PARP2, as well as Sirtuins 1, 2, 3 and 4 negatively correlated with MAP17 as shown in some datasets in Figure 5E–J.

Overall, taken together, these results indicate that MAP17 expression correlated with the increased expression of the two most important and limiting enzymes of NAD biosynthesis, NAMPT and NAPRT, which are also upregulated in pancreatic cancer. In addition, MAP17 expression correlated with a decreased expression of genes codifying for NAD-consuming enzymes, adding to the increase in NAD in pancreas tumor cells. Therefore, targeting NAD could be a potential novel therapy for pancreatic cancer.

### 3.5. MAP17-Expressing Cells Respond Better to NAMPT Inhibition Therapies and Combinations

Given the low efficacy of current treatments, we next analyzed the effect of combining gemcitabine and cisplatin, with two different NAMPT inhibitors, GMX1778 [37] and GNE617 [38], which have shown some efficacy in xenograft models [31]. We found that tumors with high levels of MAP17 were sensitive to both inhibitors. Moreover, the treatment was more effective when tumors were treated with GMX1778 or GNE617 plus gemcitabine, especially in the case of GNE617 plus gemcitabine (Figure 6). However, neither combination (GNE + GMZ or GMX + GMZ) had a greater effect than the other (Appendix A). Therefore, NAMPT inhibition decreasing the NAD pool appears to resensitize tumors to conventional therapy, achieving greater efficacy than with either the therapy or the NAMPT inhibitor alone.

## 4. Discussion

Pancreatic cancer is currently one of the most lethal malignancies, with its incidence expected to rise continuously until it is expected to become the second-leading cause of cancer-related mortality by 2030 [7,39]. It is a medically challenging cancer due to its resistance to chemotherapy and the low survival rates of patients. Treatment for localized pancreatic cancer typically involves surgery complemented by chemotherapy. Systemic treatment is crucial for metastatic pancreatic cancer as it improves both overall survival and quality of life. Nevertheless, the prognosis remains poor, with a 5-year survival rate of less than 12% [7,39]. Therefore, significant research is necessary to find more effective chemotherapy agents that induce minimal toxicity in patients. The molecular characterization of pancreatic cancer and the translation of these findings into new and targeted therapies are desperately needed for patients and represent an active area of ongoing research.

MAP17 is a small membrane protein that activates the Notch pathway [14], increases the CSC pool in tumor cells, and modifies the microenvironment to induce metastasis [15,16,40]. MAP17 overexpression has been associated with a particularly aggressive phenotype and poor prognosis in various human carcinomas, including lung, rectal, cervical, and laryngeal cancers [13,17,18,19]. In this study, we found that MAP17 was also increased in pancreatic adenocarcinoma cells and was associated with a poor prognosis, as patients with higher levels of MAP17 showed decreased survival. This alteration seems to trigger resistance to current treatment in vivo, as CSCs are proposed to be resistant to antitumoral treatments and are critical for recurrence and metastatic spread [29,41,42,43,44,45]. Therefore, genetic or microenvironmental alterations that increase the CSC pool might reduce the ability to effectively eliminate tumors, especially in pancreatic cancer.

In our study, we have observed a high MAP17 expression in pancreatic tumors, but with certain heterogeneity. This heterogeneity likely reflects the underlying biological diversity of tumors and the tumor microenvironment. Factors such as tumor subtype, tumor grade, stromal and immune cells or clonal evolution may contribute to this variability. This biological variability is consistent with the known complexity of tumor ecosystems and may influence tumor progression, therapeutic response and prognosis [16,46,47].

Furthermore, MAP17 seems to potentiate the tumorigenicity of cells [13,14,19]. In this study, we found that pancreatic cancer cells overexpressing MAP17 showed high tumorigenicity and stemness, especially in PANC-1 cells. MAP17 activates the Notch pathway, triggering a stemness program that increases the expression of genes such as HES1 and HES5, and ultimately activates core stemness genes like NANOG, KLF4, OCT4, and MYC, [14] as well as EMT-critical transcription factors [48]. Previous reports indicate that CSCs express high levels of MAP17 [40], and pancreatic cancer cells with high MAP17 levels form more tumorspheres, with a high proportion expressing CD133 antigen, a marker of cancer stem cells in multiple tumor types [49]. This phenotype has been associated with an increase in the CSC pool, leading to increased tumor malignancy. MAP17 has also been shown to induce resistance to certain treatments, such as EGFR and tyrosine kinase inhibitors, in some tumors like non-small-cell lung cancer [40]. Therefore, the enrichment of treatment-resistant CSCs may promote resistance to therapy and a poor response to treatment.

The malignant behavior induced by MAP17 has also been associated with increased ROS production mediated through SGLT1, as MAP17 inhibits Myc-induced apoptosis through ROS-mediated PI3K/AKT pathway activation [9]. Therefore, MAP17 and SGLT1 may act as predictive biomarkers to identify patients who may respond better to treatments that boost oxidative stress [12,13]. In fact, MAP17-induced sensitivity has been observed in treatments with platinum-based compounds, bortezomib, or EGFR inhibitors in lung cancer [13]. High MAP17 levels correlate with poor prognosis, and therapies that counteract MAP17 expression may be more effective in patients with few targeted-therapeutic options.

Patients with pancreatic cancer who present high levels of MAP17 exhibit worse overall survival, a finding supported by the increased resistance of MAP17-expressing tumors to classical treatments such as cisplatin, 5FU, or gemcitabine administered in vivo. In this study, we found that pancreatic cancer cells expressing high levels of MAP17 showed increased sensitivity to some drugs, such as gemcitabine, 5-fluorouracil, cisplatin, or osimertinib, an EGFR inhibitor, in vitro. However, this effect was observed only in vitro, and tumors with increased MAP17 levels did not respond to those treatments in vivo. Therefore, it is important to remark that in vitro results do not always correlate with in vivo outcomes. Tumors are very heterogeneous entities composed of different types of cells, including tumor cells, immune system cells and fibroblasts, among others. These tumors are exposed to a microenvironment that may alter the original tumorigenic potential of tumor cells. However, the tumor cell lines used in vitro, although they come from human tumors, are not in the same conditions as a tumor in a mouse and therefore do not always respond in the same way. Other factors that may reduce the efficacy of in vivo treatments could be a rapid systemic metabolism and a poor tumor penetration in the mouse model. Indeed, some studies have demonstrated that while gemcitabine has a strong cytotoxic effects in vitro, it presents a reduced efficacy in vivo [46,47]. Therefore, it is preferable to work with in vivo models, which provide a more realistic context for assessing the real efficacy of treatments. These results confirmed the difficulty of finding therapies with clear efficacy in pancreatic adenocarcinoma in vivo, even when applied in combination. Currently used clinical therapies showed only marginal responses in xenograft assays and none in MAP17-expressing xenografts.

Metabolic reprogramming is a hallmark of cancer and contributes to tumor progression and development [29]. Understanding metabolic reprogramming may facilitate the development of new therapeutic approaches. To meet the high demands of rapid proliferation, cancer cells exhibit reprogrammed metabolism, switching from the highly energy-efficient process of oxidative phosphorylation to the much less efficient process of aerobic glycolysis, known as the Warburg effect [50]. Aerobic glycolysis provides a large number of intermediate products for the biosynthesis of nucleotides, lipids, and amino acids needed to support the high proliferation rates of cancer cells [51]. Additionally, aerobic glycolysis produces lactic acid, which acidifies the tumor microenvironment, inhibiting immune cell activity and promoting tumor invasion and metastasis [51]. Under these conditions, genetic alterations that provide selective advantages may initiate and promote tumor development [52]. For instance, MAP17 overexpression increases glucose and mannose uptake through the SGLT1 transporter, activating glycolysis and increasing ROS production as a byproduct [12,53,54]. During early and later stages of tumorigenesis, impaired vascularization induces an altered microenvironment that lacks oxygen and nutrients [29]. Hypoxia has also been linked to changes in cellular metabolism because it promotes glycolysis [55]. In a study of hepatocellular carcinoma, MAP17 was described as a hypoxia-induced glycolytic regulator in the tumor microenvironment, coupling aerobic glycolysis to tumor growth via the activation of the ROS/AKT pathway and HIF1α [54]. Therefore, targeting the MAP17/ROS pathway may be an alternative approach to cancer therapy.

In addition to glycolysis, cancer cells rely on other pathways, such as glutaminolysis and serine and fatty acid synthesis, to produce macromolecules and counteract the oxidative stress caused by accelerated proliferation [56,57]. All of these pathways require the essential metabolite NAD+, which is involved in many redox and non-redox processes [31]. Cancer cells increase the NAD+ pool through enhanced biosynthesis to satisfy various cellular demands. The NAD+/NADH ratio has been found to be enhanced in cancer cells, indicating the importance of NAD in tumor metabolism [51]. In our work, we observed that pancreatic cancer cells overexpressing MAP17 exhibit increased NAD levels. The main source of NAD+ is the salvage pathway, in which NAMPT is the enzyme that catalyzes the first limiting reaction. Increased NAMPT levels have been previously associated with worse prognosis in pancreatic cancer patients [58,59] and we found that both NAMPT and NAPRT, other important enzyme involved in NAD biosynthesis, are upregulated in tumor pancreatic samples of different databases, leading to a decreased survival probability of pancreatic cancer patients. In addition, we found that MAP17 expression positively correlated with NAMPT and NAPRT expressions in different pancreatic cancer databases. Furthermore, MAP17 increased expression correlated with a decreased expression of genes codifying for NAD-consuming enzymes (such as PARP or some Sirtuins). Finally, in our cellular model, we found that tumorspheres express lower levels of SIRT1 and PARP1 (Appendix A). These data may indicate that the increase in NAD observed in pancreatic tumor cells is not only due to an increased synthesis, but perhaps more importantly, to a reduction in NAD-consuming enzymes. However, some of these correlations are modest and further functional experiments will be necessary to validate their biological significance. Therefore, our results suggest that targeting NAD could be a potential novel therapy for pancreatic cancer.

Thus, we decided to inhibit NAMPT expression in tumors with high MAP17 levels to diminish the NAD pool. We have observed that, in other tumor models, NAMPT downregulation reduces the tumorigenicity and CSC-like properties of tumor cells [31]. It makes sense to assess whether NAMPT inhibition sensitizes MAP17-expressing tumors to treatments.

In vitro, treatment with GNE617 and GMX1778, two NAMPT inhibitors, caused significant NAD pool depletion [31]. In vivo, GNE617 and GMX1778 as single treatments were more effective than cisplatin or gemcitabine (GMZ), two chemotherapy drugs currently used to treat pancreatic cancer patients. Additionally, when used in combination, GNE617 and GMX1778 treatment enhanced the antitumor effect of cytotoxic gemcitabine. NAMPT inhibition seems to sensitize tumors to conventional therapy, achieving greater efficacy with NAMPT inhibition than with either a conventional therapy or a NAMPT inhibitor alone. Therefore, NAMPT inhibitors may be used as coadjuvants with gemcitabine, the current therapy administered to pancreatic cancer patients. These findings suggest that altered NAD metabolism may contribute to the resistant phenotype and could represent a potential avenue for therapeutic exploration, but more studies are needed for further validation.

Finally, besides the relevance of the NAD pool itself, our previous findings [9,13] show that cells overexpressing MAP17 become more sensitive to treatment when exposed to both gemcitabine and the DNA-repair inhibitor olaparib, compared to control cells. In our present work, we showed that cytotoxic agents such as gemcitabine combined with NAMPT inhibitors are also synergistic. This heightened sensitivity could be the result of the decrease in DNA-repair enzyme PARPs and therefore, the accumulation of unrepaired DNA damage. These observations align with previously reported synergies between PARP inhibitors and agents like trabectedin, as well as with BRCA1/2 status and PARP1 expression in cells [60,61,62] and our own previous work. They support the idea that increasing DNA damage while inhibiting its repair may be an effective therapeutic strategy. This suggests that in tumors with a worse prognosis due to MAP17, high levels of DNA damage are present, perhaps due to MAP17-induced reactive oxygen species (ROS) that may push cells beyond a critical threshold when combined with DNA-repair inhibitors.

Given that ROS is a potent pro-apoptotic factor, contributing to DNA damage as indicated by decreasing PARPs, we propose that elevated MAP17 may enhance the effectiveness of DNA-damaging therapies combined with cytotoxic agents such as gemcitabine by tipping the balance toward apoptosis—particularly in tumors already burdened with high baseline DNA damage [9,13]. Identifying general patterns across tumor types could enable the development of therapies based on protein expression profiles rather than traditional tumor classification. Accordingly, MAP17 may represent a promising biomarker for personalized cancer treatment design.

However, our work also has some limitations. One important limitation of this study is the reduced number and the variability of the cellular models used. Some observed results showed a degree of heterogeneity and are not statistically powerful, suggesting that different and broader cellular contexts could influence the outcomes. This variability may also be explained by the activation of distinct metabolic pathways in different cell lines, which can lead to divergent responses even under similar experimental conditions.

## 5. Conclusions

MAP17 levels are elevated in pancreatic adenocarcinoma, and this overexpression correlates with decreased survival and poor prognosis. Additionally, pancreatic cancer cell lines with high MAP17 levels exhibit increased tumorigenicity and stemness, forming large, rapidly growing tumors. However, MAP17-overexpressing tumors do not respond to conventional treatments such as gemcitabine, 5-fluorouracil, or cisplatin. Given that MAP17 overexpression induces an increase in NAD production and its expression positively correlates with NAMPT expression, we treated tumors with two different NAMPT inhibitors and found that depletion of the NAD pool by these inhibitors might sensitize tumors to conventional therapy. Pancreatic tumor patients could be stratified based on MAP17 expression to receive a combination treatment of current agents with NAMPT inhibitors, which seems to sensitize cells to traditional treatments only in tumors with high MAP17 expression. Thus, our work provides the first evidence indicating that certain patients may benefit from this MAP17-based combination therapy for pancreatic cancer, where predictive biomarkers are rare and currently available targeted therapeutic options have been disappointing.

## Figures and Tables

**Figure 1 cancers-17-02575-f001:**
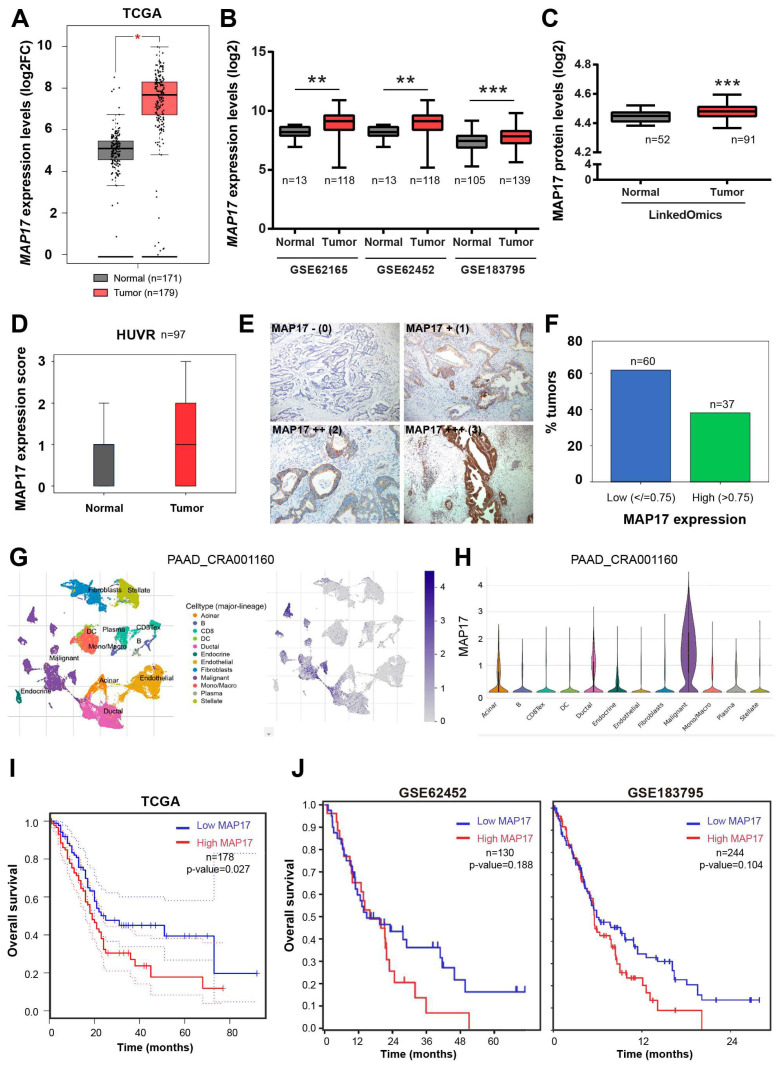
MAP17 is upregulated in pancreatic adenocarcinoma. (**A**) *MAP17* expression in a TCGA pancreatic cancer dataset. (**B**) *MAP17* expression in GSE62165, GSE62452 and GSE183795 public pancreatic cancer datasets. (**C**) MAP17 protein levels in LinkedOmics dataset. (**D**) *MAP17* expression in a HUVR-IBIS cohort. Box plots showing the expression levels of *MAP17* in pancreatic tumor tissue (red) or normal tissue (black). The data were analyzed by comparing the tumor versus the normal samples via Student’s *t* test. * *p* < 0.05, ** *p* < 0.01, *** *p* < 0.001. (**E**) Representative examples of MAP17 expression in tumors in a HUVR-IBIS cohort, as determined by immunohistochemistry. Tumor samples were classified in four categories according to the number of stained cells and the signal intensity: no MAP17 expression (0); mild MAP17 expression (1+); moderate MAP17 expression (2+); and intense MAP17 expression (3+). (**F**) Percentage of patients with low (blue) and high MAP17 (green) expression in our cohort of pancreatic tumor samples (*n* = 97). The score for high expression tumors was > 0.75. (**G**,**H**) *MAP17* expression in PAAD_CRA001160 at single-cell resolution. Graphs showing the localization of MAP17 in the pancreatic tumor microenvironment. (**I**,**J**) Kaplan–Meier plots showing the overall survival of patients with high (red) or low (blue) MAP17 expression levels in the TCGA (**I**) and GSE62452 and GSE183795 (**J**) pancreatic cancer datasets. Dotted lines indicate the confidence interval. The data were analyzed via log-rank test, and the associated *p* values are shown in the graphs.

**Figure 2 cancers-17-02575-f002:**
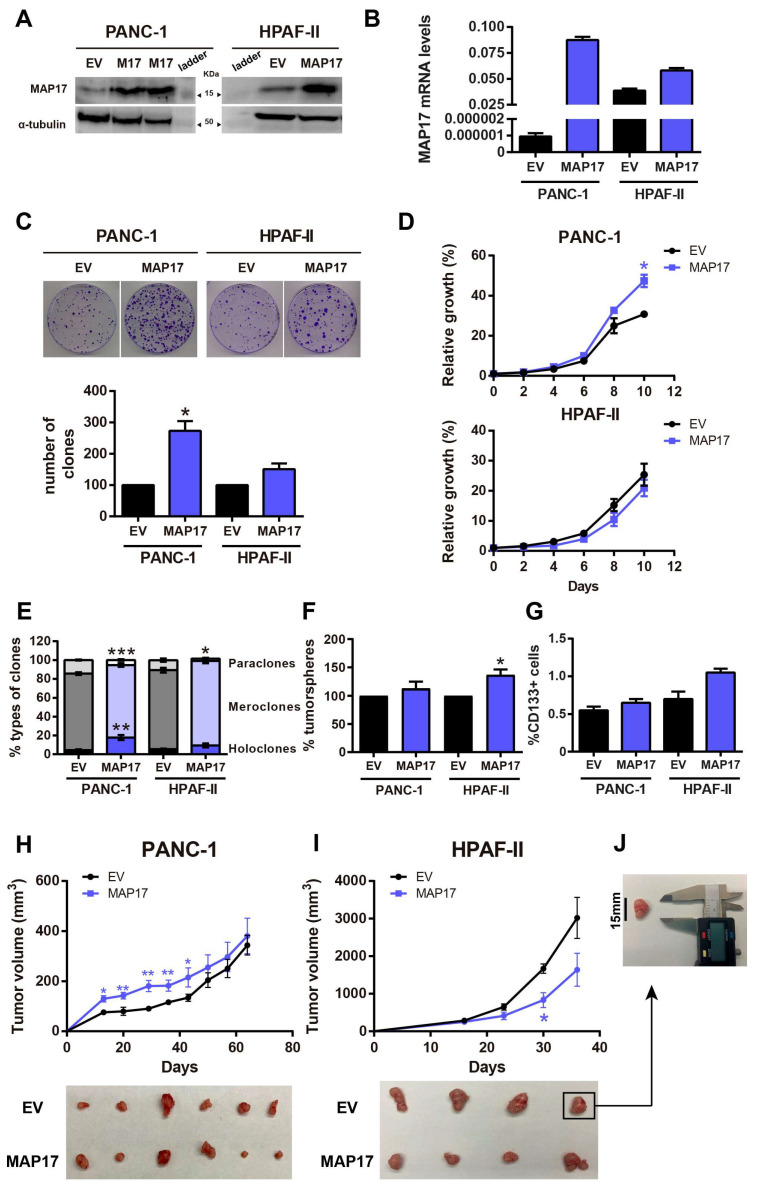
Overexpression of MAP17 increases the tumorigenicity and stemness of pancreatic cancer cell lines in vitro. (**A**,**B**) Validation of the overexpression of MAP17 in PANC-1 and HPAF-II pancreatic tumor cell lines by Western blotting (**A**) and RT–qPCR (**B**). Cells were transfected with an empty vector (EV) as a control or with MAP17 cDNA. (**C**) Clonogenic assay of PANC-1 and HPAF-II control and MAP17 cell lines. Cells were seeded at low density, and after 14 days, colonies were counted, and their sizes were measured. Representative images are shown. (**D**) Growth curves of the PANC-1 and HPAF-II control and MAP17-expressing cell lines. The values were represented referring to day 0. (**E**) Percentages of holoclones, meroclones and paraclones generated by PANC-1 and HPAF-II control and MAP17-expressing cell lines seeded at low density for 14 days. (**F**) Percentages of tumorspheres formed from PANC-1 and HPAF-II control and MAP17-expressing cell lines. (**G**) Quantification of the percentages of CD133+ cells among the PANC-1 and HPAF-II control and MAP17-expressing cells, as determined by FACS. (**H**,**I**) Growth of xenograft tumors formed from PANC-1 (**H**) and HPAF-II (**I**) control (the parental cell expressing only the EV) and MAP17-overexpressing cell lines (*N* = 6). (**J**) Representative image of a tumor with a scale ruler to visualize the sizes of the tumors. Cells were injected into nude mice, and tumor size was measured weekly. The mean ± standard deviation for a minimum of 3 independent experiments performed in triplicate is presented. Statistical analysis was performed with Student’s *t* test, * *p* < 0.05, ** *p* < 0.01, *** *p* < 0.001.

**Figure 3 cancers-17-02575-f003:**
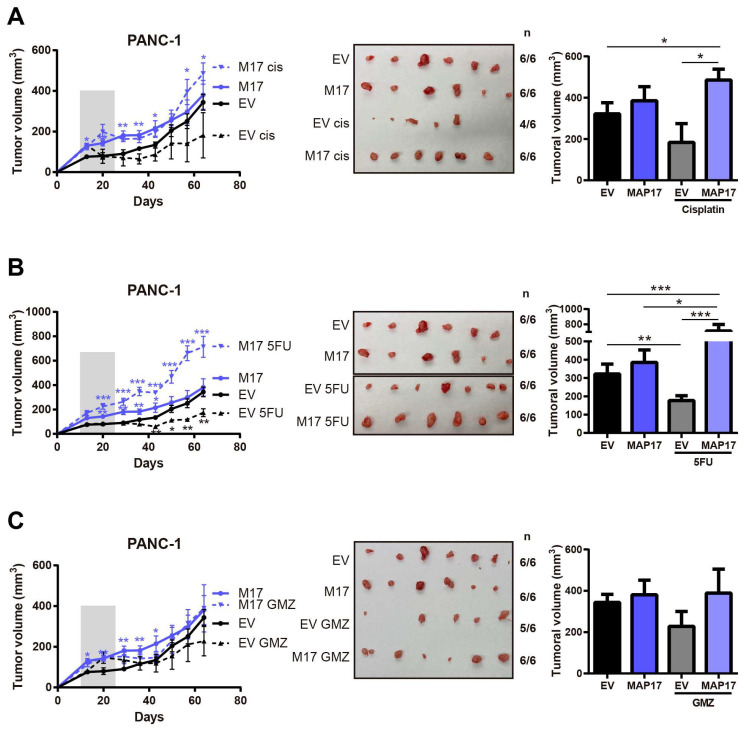
Tumors with high levels of MAP17 do not respond to conventional therapy in vivo. Determination of the tumor volume in xenograft tumors derived from PANC-1 control and MAP17-overexpressing cells (*N* = 6) after treatment with cisplatin (**A**), 5FU (**B**), gemcitabine (GMZ) (**C**). Cells were injected into nude mice, and tumor size was measured weekly. The mice received the treatment for 3 weeks. On the left we show the tumor growth and on the right the statistical comparisons between groups at the end point (final tumor volume). In the tumor growth curves, the gray bar indicates the period of treatment. Images show the sizes of the tumors at the end of the experiment. To facilitate the interpretation of the data, we showed each treatment individually, but controls are the same in each case. The mean ± standard deviation is presented for a single experiment with six independent samples (*n* = 6). Statistical analysis was performed with Student’s *t* test, * *p* < 0.05, ** *p* < 0.01, *** *p* < 0.001.

**Figure 4 cancers-17-02575-f004:**
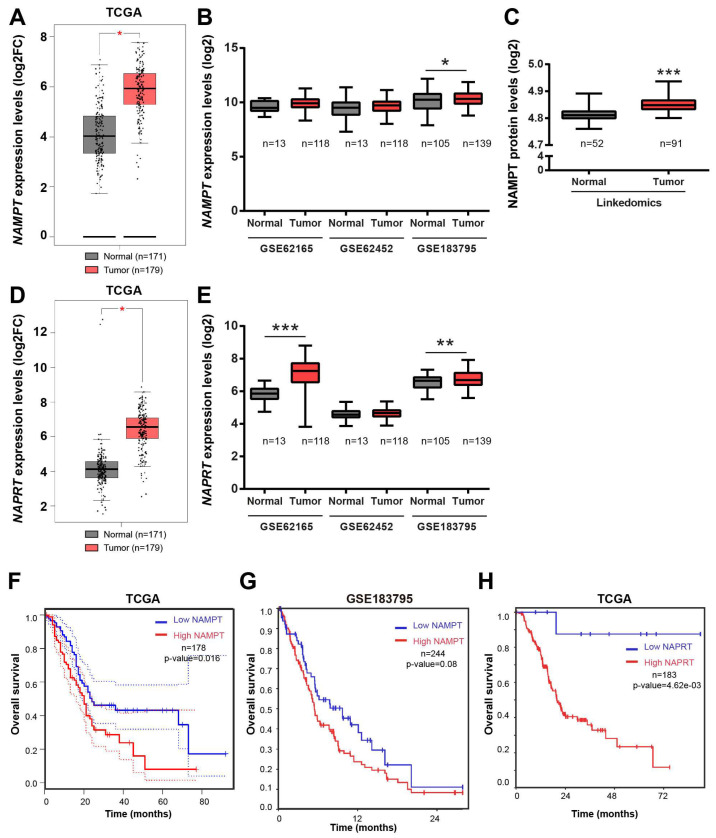
NAMPT and NAPRT are upregulated in pancreatic adenocarcinoma. (**A**) *NAMPT* expression in a TCGA pancreatic cancer dataset. (**B**) *NAMPT* expression in GSE62165, GSE62452 and GSE183795 public pancreatic cancer datasets. (**C**) NAMPT protein levels in LinkedOmics dataset. (**D**) *NAPRT* expression in a TCGA pancreatic cancer dataset. (**E**) *NAPRT* expression in GSE62165, GSE62452 and GSE183795 public pancreatic cancer datasets. (**F**,**G**) Kaplan–Meier plots showing the overall survival of patients with high (red) or low (blue) NAMPT expression levels in the TCGA (**F**) and GSE183795 (**G**) pancreatic cancer datasets. (**H**) Kaplan–Meier plots showing the overall survival of patients with high (red) or low (blue) NAPRT expression levels in the TCGA pancreatic cancer datasets. Dotted lines indicate the confidence interval. The data of survival were analyzed via log-rank test, and the associated *p* values are shown in the graphs. Statistical analysis was performed with Student’s *t* test, * *p* < 0.05, ** *p* < 0.01, *** *p* < 0.001.

**Figure 5 cancers-17-02575-f005:**
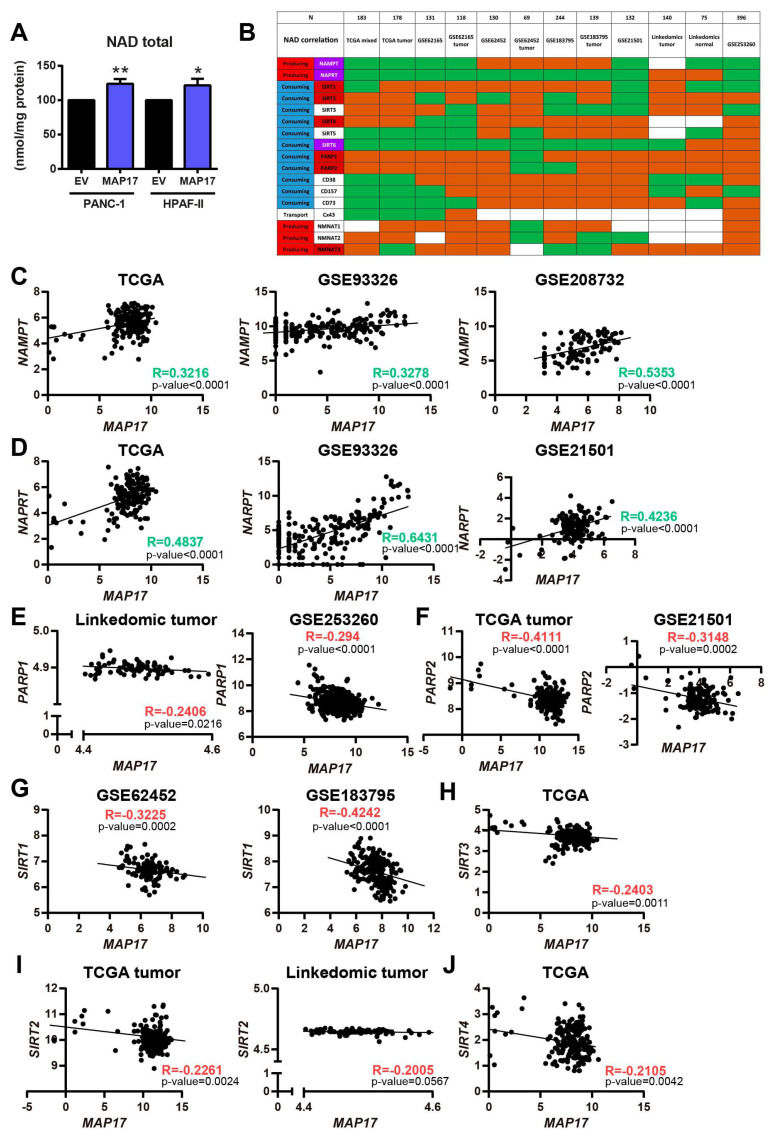
MAP17 expression correlates with NAMPT and NAPRT expressions, two enzymes involved in the NAD biosynthesis. (**A**) Total NAD levels in PANC-1 and HPAF-II control and MAP17-overexpressing cell lines as determined by cyclization assay. (**B**) Correlation of *MAP17* and different enzymes involved in NAD biosynthesis in 12 different pancreatic cancer databases. Green boxes indicate a positive correlation, whereas red boxes indicate a negative correlation. White boxes show no correlation. (**C**) Correlation of *NAMPT* and *MAP17* in TCGA, GSE93326 and GSE208732 datasets. (**D**) Correlation of *NAPRT* and *MAP17* in TCGA, GSE93326 and GSE21501 datasets. (**E**) Correlation of *PARP1* and *MAP17* in the LinkedOmics and GSE253260 datasets. (**F**) Correlation of *PARP2* and *MAP17* in the TCGA and GSE21501 datasets. (**G**) Correlation of *SIRT1* and *MAP17* in GSE62452 and GSE183795 datasets. (**H**) Correlation of *SIRT3* and *MAP17* in the TCGA dataset. (**I**) Correlation of *SIRT2* and *MAP17* in the TCGA and LinkedOmics datasets. (**J**) Correlation of *SIRT4* and *MAP17* in the TCGA dataset. Statistical analysis was performed with Student’s *t* test, * *p* < 0.05, ** *p* < 0.01.

**Figure 6 cancers-17-02575-f006:**
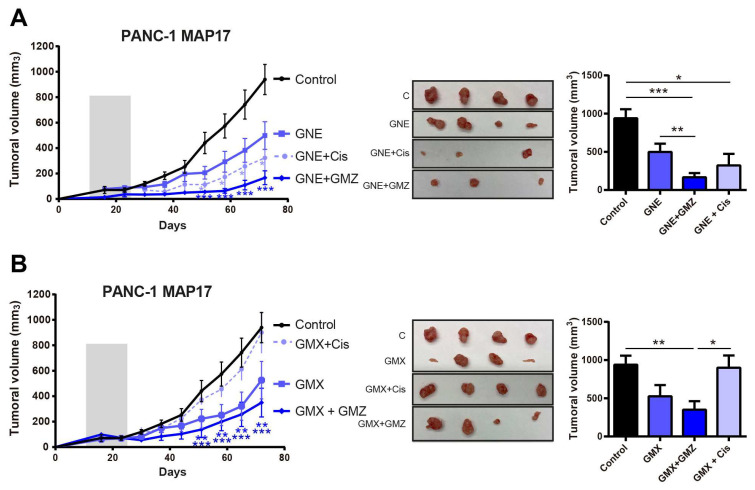
The inhibition of NAMPT sensitizes MAP17 tumors to conventional therapy in vivo. (**A**,**B**) Determination of the tumor volume in xenografts derived from PANC-1 cells that overexpressed MAP17 (*N* = 4) after treatment with GNE617 (**A**) or GMX1778 (**B**) alone and in combination with gemcitabine (GMZ) or cisplatin (Cis). Cells were injected into nude mice, and tumor size was measured weekly. Mice received the treatment for 3 weeks. Figures show the average of two different experiments performed independently. On the left we show the tumor growth and on the right the statistical comparisons between groups at the end point (final tumor volume). In the tumor growth curves, the gray bar indicates the period of treatment. Images show the sizes of the tumors at the end of the experiment. To facilitate the interpretation of the data, we showed each NAMPT treatment individually, but controls are the same in each case. The mean ± standard deviation is presented for a single or two independent experiments with four independent samples (*n* = 4) Statistical analysis was performed with Student’s *t* test, * *p* < 0.05, ** *p* < 0.01, *** *p* < 0.001.

**Table 1 cancers-17-02575-t001:** Correlation of MAP17 levels with sensitivity or resistance to antitumoral agents in vitro.

IC50 (μM)	PANC-1			HPAF-II		
	EV	MAP17	*p*-Value	EV	MAP17	*p*-Value
Gemcitabine	0.32 ± 0.09	0.09 ± 0.02	(*) 0.035	6922 ± 2265	2138 ± 337.3	(*) 0.0183
5-FU	2.42 ± 0.55	1.64 ± 0.13	0.145	501.62 ± 44.40	267.72 ± 55.31	(*) 0.0124
Cisplatin	22.92 ± 3.68	13.28 ± 1.79	(*) 0.046	68.80 ± 10.29	73.93 ± 10.33	0.733
Docetaxel	0.001 ± 0.0002	0.003 ± 0.0009	(*) 0.0292	139.02 ± 16.15	88.17 ± 6.78	(*) 0.02
Bortezomib	0.010 ± 0.0003	0.01 ± 0.001	0.555	0.013 ± 0.0062	0.02 ± 0.012	0.384
Ixazomib	0.35 ± 0.008	0.42 ± 0.03	0.054	2.4 ± 0.32	2.84 ± 0.56	0.403

The mean ± standard deviation and *p*-value for a minimum of 3 independent experiments performed in triplicate is presented. Statistical analysis was performed with Student’s *t* test, * *p* < 0.05.

## Data Availability

Data from publicly available clinical and genomic information were obtained from the R2 Genomics analysis and visualization platform (https://hgserver1.amc.nl/cgi-bin/r2/main.cgi; accessed on January 2025), the TCGA Research Network (https://cancergenome.nih.gov/; accessed on November 2022) and the TISCH resource (http://tisch1.comp-genomics.org/; accessed on November 2022) [25].

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
