# Peer review of "Efficacy of NAMPT Inhibitors in Pancreatic Cancer After Stratification by MAP17 (PDZK1IP1) Levels"

_cancers, 2025, doi:10.3390/cancers17152575_

Round 1
Reviewer 1 Report
Comments and Suggestions for Authors
Dear Authors
Your paper is interesting and may add a new therapeutic approach to a difficult disease such as pancreatic cancer.
There are minor issues that I added in the attached file.

Author Response
We thank this reviewer for the positive comments. We have reviewed the manuscript taking into consideration all of the comments and suggestions. Our reply is in the attached document.

Reviewer 2 Report
Comments and Suggestions for Authors
The manuscript by Verdugo Sivianes et al. investigates the prognostic and predictive significance of MAP17 in pancreatic cancer. The authors correlate MAP17 expression with response to standard chemotherapies and NAMPT inhibitors. The authors demonstrate that elevated MAP17 levels are associated with a worse prognosis in pancreatic cancer. However, there are significant concerns regarding the robustness of the findings for translation into clinical practice and the depth of the mechanistic analysis.
My main concern is the divergent responses to MAP17 overexpression observed in the two pancreatic cancer cell lines used. This variability raises concerns about cell-line-specific effects. To validate the generalisability of MAP17’s impact on drug sensitivity and NAD metabolism, the authors should include a third, mechanistically distinct pancreatic cancer cell line.
Furthermore, a critical limitation is the absence of MAP17 loss-of-function experiments. The current conclusions rely on overexpression models, which are divergent in themselves. Including knockdown/knockout studies in both in vitro and in vivo settings would strengthen the conclusions.
The manuscript claims that MAP17 is both a prognostic and predictive biomarker. However, given the inconsistent treatment responses and lack of functional confirmation, predictive claims should be made cautiously.
Author Response
We thank the reviewer for his/her time in reviewing our manuscript. We understand his/her concerns about some aspects of our work. Our reply is in the attached document.

Reviewer 3 Report
Comments and Suggestions for Authors
- This study investigates the role of MAP17 in pancreatic cancer and its implications for treatment stratification—a topic that is both clinically relevant and underexplored. Your findings, especially the suggestion that NAMPT inhibition may enhance gemcitabine sensitivity in MAP17-high tumors, are intriguing.
That said, the presentation could be improved. In particular, the Results section includes too many speculative statements and interpretive remarks, which may distract from the core findings. These would be more appropriately discussed in the Discussion. Clarifying the separation between objective results and interpretation would improve both clarity and rigor.
With some restructuring and careful revision, this study has the potential to make a strong impact.

To increase clarity and readability, manuscripts may benefit from careful English editing, with particular attention to maintaining consistent verb tenses, avoiding ambiguous phrases (e.g., “in some respect”), and adhering to a formal academic tone.
Author Response
We thank the reviewer for his/her time in reviewing our manuscript. We have reviewed the manuscript taking into consideration all of the comments and suggestions. We have taken special care to differentiate objective results and discussion, as recommended. Our reply is in the attached document.

Reviewer 4 Report
Comments and Suggestions for Authors
The present article evaluated the role of MAP17 in the tumorigenesis and drug resistance of pancreatic cancer and identified NAMPT as one of the mediators. The authors used PDAC cells in vitro and in vivo xenograft models to determine the role of MAP17 overexpression in PDAC tumorigenesis and drug resistance and utilized two NAMPT inhibitors to override the detrimental effects of MAP17 overexpression. Although the premise of the study is interesting, several major concerns were identified.
- Median survivals were not much different in TCGA and are exactly the same in both GSE databases (Fig. 1I-J).
- It is not clear how the different types of clones were identified in Fig. 2E-F. Please add the details in the methods section.
- Although MAP17-overexpressing PANC-1 cells showed an initial increase in tumor volume compared to EV group, the curves meet after some time (Fig. 2H). This is not due to reaching plateau in the MAP17 group, as both the groups are still in the growth phase. Did the authors check MAP17 expression at endpoint tumors to confirm that MAP17 overexpression is still there or not?
- It is interesting to note that MAP17-overexpressed HPAF-II led to significantly more tumorspheres (Fig. 2F), but lower tumor volume than EV (Fig. 2I). Shouldn’t they show similar trends? The authors reasoned that this may be caused by an increase in non-fast cycling CSC, but they did not show any proof to validate this line of reasoning.
- Please also provide significance values for these correlation plots shown in Fig. 5C-J.
- GNE group in Fig. 6A, and GMX and GMX+GMZ groups in Fig. 6B showed high intragroup variability in tumor volumes, with ~50% of the tumors in the treatment groups being more or less equal to that of the control groups. While this suggests probable responders and non-responders within the treatment groups, this is not usually seen in xenograft studies since they are not as heterogenous as patient populations. The authors should increase the number of mice per group to confirm the efficacy of GNE or GMX in MAP17-overexpressing PANC-1 xenografts.
- Although GMX_GMZ showed slightly better antitumor effect than GMX in PANC1- xenografts, the GMX+Cis led to increased tumor volume than GMX alone (Fig. 6B). The authors did not elaborate on this striking difference. Also, GMX+Cis tumors in the manuscript (Fig. 6B) do not match the original figure as shown in the supplementary document. Based on the supplementary document, it seems GMX+Cis group tumors are smaller than control group, however, this does not reflect in Fig. 6B in the manuscript.
Author Response
We thank the reviewer for his/her time in reviewing our manuscript. We have reviewed the manuscript taking into consideration all of the comments and suggestions. Our reply is in the attached document.

Round 2
Reviewer 2 Report
Comments and Suggestions for Authors
The authors have adequately addressed the main concern raised in the previous review, and the additions made to the manuscript are appropriate and sufficient.
Author Response
We thank this reviewer for the positive comments, we are glad we responded appropriately.
Reviewer 4 Report
Comments and Suggestions for Authors
I appreciate the authors responding to my concerns; however, I am not satisfied with some of their responses:
- It is not clear how the authors determined the percentage of CSCs in the colonies to classify them holoclone, meroclone, or paraclone. Did they check any specific CSC marker in those clones or was their inference based on morphology only (i.e. compact vs irregular)? The methods section should describe this clearly.
- In response to my comment about the discrepancy of GMX+Cis tumor images between Fig. 6 and supplementary data, the authors mentioned that ‘There was an error with the scale of the photos in the supplementary material that we have already corrected, so now Figure 6B corresponds to the supplementary original image. We apologize for any confusion we may have caused in the reviewer.’ However, to me, the new supplementary images show a completely different set of GMX+Cis tumor images than the previously presented supplementary figure in prior submission. To remove the confusion, the authors should provide all the tumor images from 3 independent experiments, at least for review if not for publication.
Author Response
We thank the reviewer for his/her time in reviewing our manuscript. We regret not having answered some questions appropriately so we will try to do it better. Our reply is in the attached document.
